# Accuracy of Resting Metabolic Rate Prediction Equations in Sport Climbers

**DOI:** 10.3390/ijerph20054216

**Published:** 2023-02-27

**Authors:** Anna Chmielewska, Krzysztof Kujawa, Bożena Regulska-Ilow

**Affiliations:** 1Department of Dietetics and Bromatology, Wrocław Medical University, 50-367 Wrocław, Poland; 2Statistical Analysis Centre, Wrocław Medical University, 50-367 Wrocław, Poland

**Keywords:** basal metabolism, sport climbing, indirect calorimetry

## Abstract

Resting metabolic rate (RMR) represents the energy required to maintain vital body functions. In dietary practice, RMR is determined by predictive equations on the basis of using body weight or fat-free mass. Our study aimed to assess whether predictive equations used to estimate RMR are reliable tools for estimating the energy requirements of sport climbers. The study included 114 sport climbers whose RMR was measured with a Fitmate WM. Anthropometric measurements were performed with X-CONTACT 356. The resting metabolic rate was measured by indirect calorimetry and was compared with the RMR estimated by 14 predictive equations on the basis of using body weight/fat-free mass. All equations underestimated RMR in male and female climbers, except for De Lorenzo’s equation in the group of women. The De Lorenzo equation demonstrated the highest correlation with RMR in both groups. The results of the Bland–Altman tests revealed an increasing measurement error with increasing metabolism for most of the predictive equations in male and female climbers. All equations had low measurement reliability according to the intraclass correlation coefficient. Compared with the indirect calorimetry measurement results, none of the studied predictive equations demonstrated high reliability. There is a need to develop a highly reliable predictive equation to estimate RMR in sport climbers.

## 1. Introduction

Resting metabolic rate (RMR) is defined as the energy required to maintain vital bodily functions, such as energy substrate metabolism, respiration, body temperature and heart rate while the body is at rest [1]. It is estimated that in healthy individuals characterized by a low level of physical activity, RMR accounts for 60–70% of the total daily energy expenditure [2]. Resting metabolic rate can be influenced by age, endocrine function, the dietary energy value and the activity of the nervous system [3]. Body composition, particularly fat-free mass (FFM), can significantly affect individual differences in RMR, as confirmed by the study conducted in untrained healthy individuals [4].

It has been suggested that RMR can be increased by resistance [5] and interval training [6]. Indirect calorimetry (IC) is considered the gold standard for determining RMR [7]. However, IC measurements are expensive and labor-intensive, which limits their use in clinical practice [8]. Additionally, they have been validated in different study groups, which may reveal many errors if applied to a specific population [8].

Recently, sport climbing has become increasingly popular. Since 2020, it has been included in the Olympic program and is carried out in three categories: bouldering, lead climbing and speed climbing. Climbers are characterized by a specific body type. A low percentage of body fat and low body mass are considered determinants of their climbing success [9]. Along with the increased interest in climbing, there have been many studies on sport climbers, including their nutritional habits. The energy requirements of climbers are calculated with commonly used predictive equations developed for other athletes [10,11], which may lead to some discrepancies between the obtained RMR results and the actual RMR values. To date, studies have focused on estimating the accuracy of selected predictive equations to calculate RMR in various athletes [12,13].

Recently, more and more studies have evaluated energy availability across the representants of sport climbers because the representants of this discipline are suggested to be in danger of eating disorders resulting from the desire to lower their body weight [14,15]. Resting energy expenditure is used as a parameter to define energy deficiency among sport participants [16]. To assess energy needs, obtaining the proper estimation of resting metabolic rate is essential. It has been suggested that indirect calorimetry is the best method to obtain the most accurate results, and the use of available prediction equations comes with the potential to under- or overestimate this value [15].

Correct RMR estimates are the basis for developing a personalized nutritional strategy, which contributes to athletic success. Identifying a predictive equation to precisely estimate RMR in sport climbers would help coaches and sports nutritionists work with climbers.

Our study aimed to compare the accuracy of RMR calculated with 14 predictive equations, where RMR was measured by IC, and to identify the predictive equation with the highest correlation with the IC measurement results.

## 2. Materials and Methods

### 2.1. Participants

The study included 114 regular sport climbers (43 women; 71 men) from Wrocław, Poland. The criterion for inclusion in the study was at least 1 year of regular sport climbing. The exclusion criterion was recreational climbing with less than once-a-week climbing practice. Information on the study was spread through social media and directly through trainers at 5 climbing gyms. All climbers who were willing to take part for the duration of the study (2019–2021, with an obligatory break during the COVID-19 pandemic) and who fulfilled the criteria were accepted. The study was conducted in accordance with the Declaration of Helsinki and approved by the Institutional Ethics Committee of Wrocław Medical University (number KB-45/2019).

### 2.2. Study Protocol

In order to determine the energy needs of the study participants, their resting metabolic rates were measured using indirect calorimetry. Resting metabolic rate was measured by IC with Fitmate WM (Cosmed, Rome, Italy) device. The study participants had been asked to come to the study after fasting, after an overnight rest and after avoiding intense physical activity an evening before measurements, which were taken in a darkened and soundproof room. The entire test took 12 min. In the first 2 min, we matched the participants’ respective breathing patterns to the measurement requirements. The actual RMR measurement took 10 min. Anthropometric measurements, such as body weight, body fat and fat-free mass, were performed with X-CONTACT 356 (Jawon Medical Co., Seoul, Republic of Korea) analyzer and a TANITA HR-001 (TANITA, Tokyo, Japan) stadiometer. Contraction force was measured with MAP 80K1 hand-grip dynamometer (KERN & Sohn GmbH, Balingen, Germany) to assess grip strength to body mass ratio (SMR). Systolic and diastolic pressure and pulse were measured using a sphygmomanometer in order to exclude performing an indirect calorimetry in a state of anxiety. High blood pressure and pulse values could falsely increase the results.

In order to obtain the characteristic of the group and the level of advancement in climbing, participants filled out the questionnaire with the climbing grade section questions. Participants’ respective climbing grades were subjectively determined by assessing 3 routes established over the preceding 6 months. Self-reported climbing ability was considered as a valid representation of actual climbing ability [17]. Participants were asked to indicate the highest grade, lead or boulder that they had managed to redpoint on 3 routes/problems, either on an artificial wall or on a rock. The grade was then standardized according to the International Rock-Climbing Research Association (IRCRA) scale [18]. According to the declared climbing grade, we identified 4 levels of advancement groups: beginners (4 participants), intermediate (47 participants), advanced (51 participants) and elite (12 participants).

### 2.3. Equations for Analyzed Resting Metabolic Rate (RMR)

The obtained results were compared to RMR results, estimated with 14 predictive equations. Resting metabolic rate was estimated with predictive equations on the basis of body weight, according to Harris–Benedict [3], Mifflin–St. Jeor [10], WHO/Schofield [19], Owen [20,21] Roza–Shigal [22], De Lorenzo [11], Bernstein [23], ten Haaf [20] and Tinsley [19], and on the basis of fat-free mass, according to Mifflin–St. Jeor [10], Cunningham 1980 [10], Cunningham 1991 [4], ten Haaf [19] and Tinsley [10]. Formulas for the analyzed predictive equations are presented in the Table 1.

### 2.4. Statistical Analysis

Statistical analyses were performed with STATISTICA v. 13.3 (TIBCO Software Inc., Palo Alto, CA, USA) and by using the R-package “psych” [24]. An analysis of the differences between the measured and calculated metabolic rates was separately performed for each gender. The Shapiro–Wilk test was used to assess the normality of the distribution of the studied data. For the variables whose differences were normally distributed, we used the paired sample *t*-test for related pairs. For the differences distributed nonnormally, we used the Wilcoxon signed-rank test. An intraclass correlation coefficient (ICC) was calculated for each equation to assess the correlation of each measurement, where RMR was measured with IC. The ICC values (mode “single fixed raters”) and their statistical significance were calculated with the use of function “ICC” (of the package “psych”). The Bland–Altman graphs were used to separately visualize the relationships between the two compared metabolic rates (measured vs. calculated) for each gender.

## 3. Results

Table 2 shows characteristics of the study group according to gender. Average age of participants was similar. The female group was characterized by a lower BMI.

Figure 1 shows the percentage of results obtained with the predictive equations that were within ±10% agreement with the IC measurement results.

In the group of women, the De Lorenzo and ten Haaf FFM equations demonstrated the highest correlation with the actual RMR, and in the group of men, the highest correlation was found for the De Lorenzo equation, followed by the ten Haaf FFM and ten Haaf BW equations. In both groups, Bernstein’s equation demonstrated the lowest correlation.

Table 3 shows the results of difference tests for the studied equations.

In the group of women, we obtained the smallest mean difference with the ten Haaf FFM equation, and in the group of men, with the ten Haaf BW equation. The largest differences between the means in both groups were observed with the Bernstein’s equation. In the group of women, only the De Lorenzo equation overestimated RMR. The remaining equations underestimated RMR in the female and male groups. The intraclass correlation coefficient for all equations was very low, indicating very poor correlation between RMR measured with predictive equations and by IC.

The majority of predictive equations in the Bland–Altman tests that are presented in Figure 2 demonstrated a high correlation coefficient, which implied a steadily increasing predictive equation error when RMR increased. In the group of women, we obtained favorable results with the De Lorenzo equation, which showed a small scatter of results and a difference between the mean measurement and the equation result (−60 kcal), as well as a low correlation coefficient (R = 0.33). Among equations with a small difference, Bland–Altman tests demonstrated a large scatter of results, even for the equations for which ±10% RMR correlation was about 70%, and ICC indicated their low reliability.

## 4. Discussion

Our study aimed to evaluate the accuracy of several predictive equations used to estimate RMR in a group of sport climbers compared to the RMR results measured by IC with a Fitmate WM. 

In our study, the mean values of anthropometric measurements were similar to those involving sport climbers in other studies analyzing the body composition of sport climbing representants [14,15]. Compared with the group of advanced and elite climbers in Monedero et al. [14], the elite climbers in Gibson-Smith [15], and the Polish advanced climbers in Sas-Novosielki et al. [25], the male group was characterized with a higher body weight and fat percentage, but in the female group, the characteristic of the group was convergent. These differences may have been due to the different climbing levels of the study participants, while in the abovementioned studies, the specific characteristics were due to the study of a small group of top-level climbers. Furthermore, the group in our study was larger and included representatives at various levels of advancement.

Many commonly used equations to predict RMR are based on body weight (BW) alone, without considering muscle mass or fat-free mass (FFM) [26]. People of the same body weight may present significantly different body compositions, which may affect RMR. Given the important role of muscle tissue in determining energy requirements, it seems that in a population of physically active individuals, predictive equations based on fat-free mass provide more-accurate results. Predictive equations with the highest correlation with IC results in a group of inactive, normal weight and obese individuals can result in very different correlation outcomes in different athletes.

Frankfield et al. [27] found that among all equations used in dietary practice, the Mifflin–St. Jeor equation (based on actual BW) demonstrated the highest correlation with RMR. In our study, this equation was 28% consistent with RMR obtained by IC among female and male sport climbers, and its results underestimated RMR on average as being 235.29 and 301.03 kcal (R = 0.51; *p* < 0.001; R = 0.64; *p* < 0.01).

In the study by Jagim et al. [12], the Cunningham equation was the most accurate for estimating RMR in female athletes. However, among male study participants, the Harris–Benedict equation demonstrated the highest correlation with IC measurements. In our study, the De Lorenzo equation demonstrated the highest correlation with IC measurements for both men and women: respectively, 93% and 70% of the results of predictive equations correlated 90–110% of the time with IC results. The De Lorenzo equation overestimated RMR on average by 60.68 kcal in a group of women and underestimated RMR by about 113.11 kcal in a group of men. Among women, the 1980 Cunningham equation demonstrated a high correlation with RMR (77% concordant results, average difference of 65.07 kcal). The ten Haaf equation, based on fat-free mass, demonstrated a 79% agreement with RMR, with a mean difference of 46.73 kcal, which represented the smallest difference between the average results. However, the high correlation coefficient in the Bland–Altman test demonstrated its low usefulness, because the results differed with the increasing RMR (R = 0.47, *p* < 0.001). In the group of men, the ten Haaf equations, based on FFM and BW, demonstrated an identical correlation (68%) (134.94 and 98.5 kcal difference). These findings are convergent with the latest systematic review of the accuracy of the RMR predictive equations in the group of athletes conducted by Marthino et al. [16], where the De Lorenzo and ten Haaf equations were observed to have the highest compatibility. In the group of recreational athletes aged 18–35 years, RMR obtained with the Cunningham equation was the most consistent with the results obtained by IC. The De Lorenzo equation demonstrated slightly lower agreement, while RMR obtained with the Harris–Benedict, Schofield, Owen and FAO equations were characterized by low accuracy [20].

In our study, RMR obtained with the Schofield and Owen equations demonstrated low accuracy for both men and women. The Harris–Benedict equation overlapped with the IC results by 44% in the group of men and by 56% in the group of women. The mean differences were 252.62 and 67.75 kcal, respectively, but in the group of women, the mean difference was not statistically significant. Similarly, in our study, the Cunningham equation demonstrated a high correlation with RMR in the group of women and a lower correlation in the group of men.

Frings-Meuthen et al. [28], who studied masters athletes, found that the Harris–Benedict, WHO and Muller equations underestimated RMR and that the Cunningham equation overestimated it. The authors obtained the most precise results with the De Lorenzo equation, which overestimated RMR only in the group of women. Other studied equations underestimated RMR by 46 kcal or by even 588 kcal.

Mackay et al. [29], who assessed the accuracy of the Harris–Benedict, Mifflin and WHO predictive equations in female athletes at the recreational and subelite levels, demonstrated their low correlation with IC measurements. The authors also assessed whether RMR measurements correlated with RMR estimated with predictive equations, depending on the measurement duration. The intraclass correlation coefficient (ICC) was the highest for the analyzed equations when the measurement took longer than 15 min, which may indicate a need for a longer measurement to more accurately calculate RMR. In our study, the measurement took 10 min, and the ICC values for all equations were low. We found significant differences between the measurements results after dividing the study group, according to the IRCRA scale, into intermediate, advanced and elite climbers, which may suggest that dividing the study group according to climbing sophistication would help obtain more-precise results.

The correct estimation of RMR is also crucial in preventing the onset of reduced energy availability, which can lead to relative energy deficiency in sport (RED-S) in male and female athletes [13,30]. Given the fact that energy-restricted diets are popular among sport climbers [31], they are at risk of developing micronutrient deficiencies and the insufficient availability of energy, which can lead to endocrine and metabolic disorders [32]. Additionally, endocrine disorders themselves can affect RMR, so preventing insufficient energy supply can help maintain RMR within the total daily energy expenditure. It has been estimated that an RMR ratio < 0.9 may be a suitable indicator for assessing RED-S [32]; however, to be accurately used, it requires properly developed equations. Further, in our study, the average body weight of sport climbers was lower compared to that of the athletes analyzed in other studies [3,12,19,20,29]. The nature of sport climbing molds individuals into this body type. Climbers strive to have a low body weight and a low percentage of body fat, which affects their actual RMR. Similarly to previous studies carried out on athletes, we observed that predictive equations tended to underestimate RMR values [12,30].

Future research should aim to develop sport-specific RMR prediction equations to account for large differences in body composition and training levels commonly observed in different athletes [3]. With the growing popularity of sport climbing and its inclusion in the Olympic program in 2020, it seems important to develop an appropriate tool for estimating RMR in sport climbers. To date, no predictive equations have been developed specifically for predicting RMR in sport climbers whose body type significantly differs from other athletes, which may explain the discrepancies between the RMR results estimated with predictive equations and the RMR results measured by IC. The RMR predictive equations developed so far have been validated with different representants from various sport climbing populations: overweight individuals, nonactive individuals and athletes of different disciplines. Using them to predict the RMR of sport climbers is not an effective method to plan nutritional strategies on the basis of energy needs.

Properly assessing athletes’ energy requirements is an important element of their collaboration with sports nutritionists. A precise tool to enable that will allow the adjustment of energy supply according to the demand, resulting from the training undertaken, and will prevent unexpected changes in body weight due to RMR underestimation. On the other hand, there are various aspects that should be considered when developing predictive equations for sport representatives. Depending on the type of climbing preferred, such as bouldering, rope climbing and speed climbing, sport climbers may differ in their anthropometric measurements [33]. In addition, the athletes’ body compositions can fluctuate depending on the season, which were also observed in other sport disciplines [34]. These factors undoubtedly make the development of a sport-specific formula more difficult. Although it is not in practice possible to conduct indirect calorimetry, it is important to develop an equation that is based on many measurements taken among representatives of various types of climbing, to take into account training macrocycles.

## 5. Limitations of the Study

The study has some limitations that should be mentioned. The analyzed group is characterized by various levels of advancement. As specific body anthropometric measurements, such as small body mass, low body fat and high fat-free mass, are characteristic of advanced and elite climbers, such values may not be represented by intermediates and beginners. A wide range of these components could influence the results. Additionally, a high fat-free mass value influences the resting metabolic rate, and the obtained results of Bland–Altman plots show that the higher the RMR values, the bigger the error, according to the results of the formula.

To obtain the most accurate results from the indirect calorimetry test, it should be carried out for at least 30–40 min. In our study, this period was 10 min, with additional 2 min to match the participants’ respective breathing patterns to the measurement requirements. To obtain the most accurate results that could be used for developing a reliable RMR prediction for this discipline, as mentioned in the discussion, the results of different types of climbing representatives and various training macrocycles should be considered.

## 6. Conclusions

None of the analyzed predictive equations is a reliable tool for estimating resting metabolic rate in both female and male representants of sport climbing. Developing a highly reliable predictive equation, one that is accurate for sport climbers in terms of their specific body compositions, seems to be an essential element in sports dietetics and a key factor to help athletes build their nutritional habits, which are an important component in their performance success.

## Figures and Tables

**Figure 1 ijerph-20-04216-f001:**
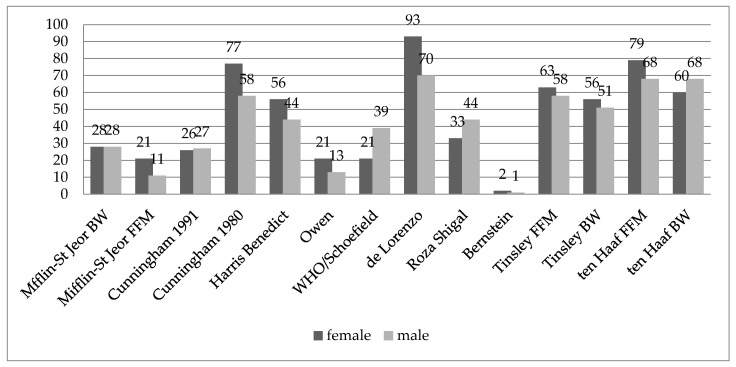
Percentage of studied equations overlapping by ±10% with IC measurement results for female group and male group.

**Figure 2 ijerph-20-04216-f002:**
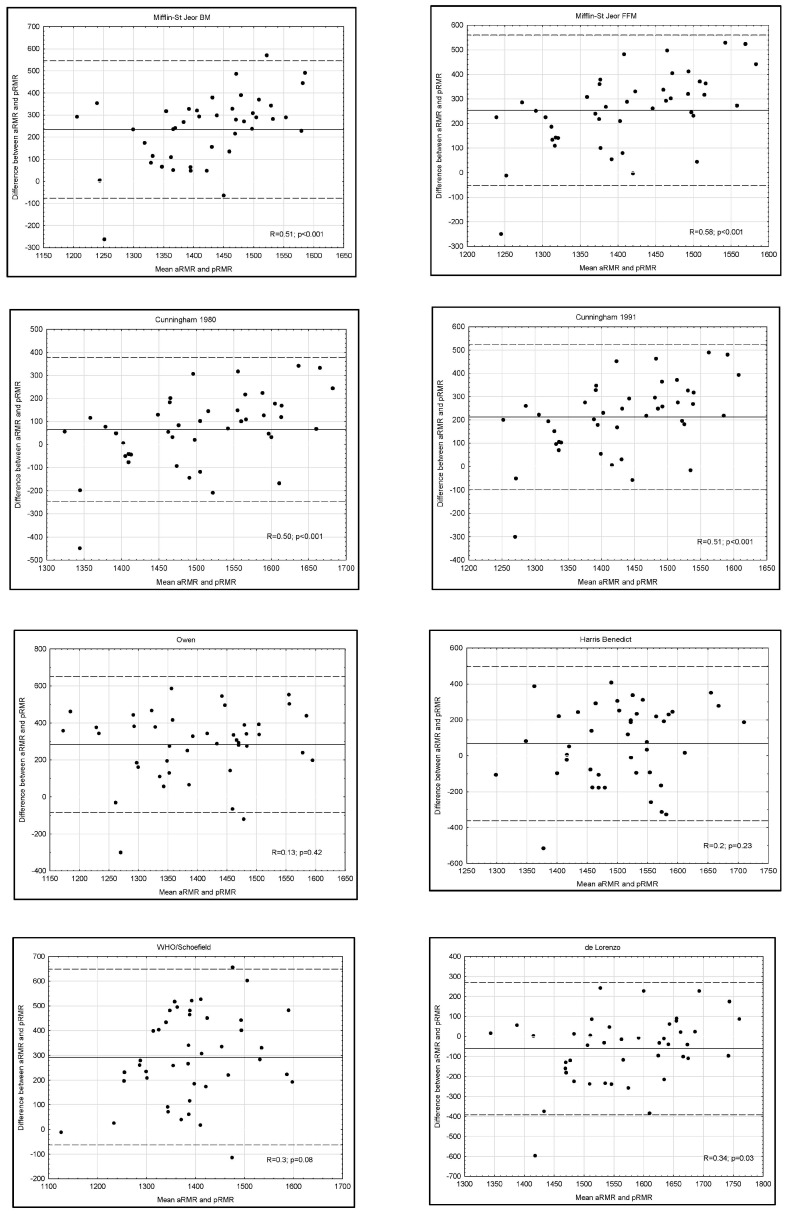
Bland–Altman plots for IC-RMR and predictive RMR equations for the subjects. The solid line represents the mean difference (BIAS) between RMR measured by Fitmate WM and predicted RMR. The upper and lower dashed lines represent the 95% limits of agreement. Here, (**A**) represents Bland–Altman plots of female athletes and (**B**) represents Bland–Altman plots of male athletes.

**Table 1 ijerph-20-04216-t001:** Resting metabolic rate (RMR) (kcal/day) equations considered in the study.

Equations	RMR Predictive Equations Formula
**Mifflin–St Jeor BW**	9.99 ∗ BW + 6.25 ∗ H − 4.92 ∗ A + 166 ∗ gender − 161
**Mifflin–St Jeor FFM**	19.7 ∗ FFM + 413
**Cunningham 1991**	21.6 ∗ FFM + 370
**Cunningham 1980**	22 ∗ FFM + 500
**Harris–Benedict**	M: 13.75 ∗ BW + 5 ∗ H – 6.76 ∗ A + 66.47 F: 9.56 ∗ BW + 185 ∗ H − 4.68 ∗ A + 655.1
**Owen**	M: 879 + 10.2 ∗ BW F: 50.3 + 21.6 ∗ BW
**WHO/Schoefield**	M: 18–30 y = 15.057 × W + 692.2 30–60 y = 11.472 × W + 873.1 F: 18–30 y = 14.818 × W + 486.6 30–60 y = 8.126 × W + 845.6
**De Lorenzo**	9 ∗ BW + 11.7 + H−857
**Roza–Shigal**	M: 88.362 + (13.397 ∗ BW) + (4.799 ∗ H) − (5.677 ∗ A) F: 447.593 + (9.247 ∗ BW) + (3.098 ∗ H) − (4.33 ∗ A)
**Bernstein**	M: 11.02 ∗ BW + 10.23 ∗ H – 5.8 ∗ A − 1032 F: 7.48 ∗ BW + 0.42 ∗ H − 3 ∗ A + 844
**Tinsley FFM**	25.9 ∗ FFM + 284
**Tinsley BW**	24.8 ∗ BW + 10
**ten Haaf FFM**	0.239 (995.272 ∗ FFM + 284)
**ten Haaf BW**	0.239(49.94 ∗ BW + 24.59 + H − 34.014 ∗ A + 799.257 ∗ gender + 122.502)

BW—body weight (kg), H—height (cm), FFM—fat-free mass (kg), A—age (y), gender (female: 0, male: 1).

**Table 2 ijerph-20-04216-t002:** Characteristics of the study group.

Measured Parameter	Female (n = 43)		Male (n = 71)	
	x ± SD	Min–Max	x ± SD	Min–Max
**Age (y)**	30 ± 7.1	21–53	30.6 ± 6.2	18–47
**Body mass (kg)**	57.2 ± 6.3	42.8–72.5	72.4 ± 7.7	58.9–92.3
**Height (cm)**	166 ± 5.3	153–175	178,1 ± 6.4	166.5–197
**Fat-free mass (kg)**	44.3 ± 4	36.2–54.3	60.3 ± 6	49.4–72.3
**Body fat (%)**	22 ± 4.7	13.1–32.6	16.4 ± 4.5	5.5–31.5
**Body fat (kg)**	13.4 ± 5.5	6.4–39.2	12.6 ± 6.5	4.2–53.6
**BMI (kg/m^2^)**	20.7 ± 1.8	17.1–25.7	22.7 ± 1.9	18.5–29.0
**Contraction force right (kg)**	32.8 ± 5.8	17.5–49.8	48.7 ± 7.8	27.5–70.1
**Contraction force left (kg)**	29.3 ± 5.9	16–42.7	45.2 ± 7.8	19.3–66.8
**Systolic blood pressure (mmHg)**	104.3 ± 12.1	69–146	122.7 ± 12.6	103–165
**Diastolic blood pressure (mmHg)**	66.9 ± 10.5	34–97	73.8 ± 9.8	58–114
**Pulse (bpm)**	64.3 ± 10.3	44–87	63.2 ± 11	45–101
**RMR (kcal)**	1539.16 ± 152	1120–1831	1992 ± 219.7	1542–2714

x—mean; SD—standard deviation.

**Table 3 ijerph-20-04216-t003:** Comparison of the differences between the results calculated with predictive equations and by ICC. Here, x = RMR – C, where C is the calculated metabolic rate.

Equation	Female (n = 43)					Male (n = 71)				
	ICC	x ± SD	*p*	CI − 95	CI + 95	ICC	x ± SD	*p*	CI − 95	CI + 95
**Mifflin–St Jeor BW**	0.18	235.29 ± 158.87	0.000 *	186.40	284.19	0.29	301.03 ± 207.01	0.000 *	252.04	350.03
**Mifflin–St Jeor FFM**	0.17	253.91 ± 156.26	0.000 *	205.82	302.00	0.26	391.28 ± 214.42	0.000 *	340.53	442.04
**Cunningham 1991**	0.18	212.79 ± 158.78	0.000 *	163.92	261.65	0.28	319.73 ± 217.33	0.000 *	268.29	371.17
**Cunningham 1980**	0.18	65.07 ± 159.36	0.011	16.03	114.12	0.28	165.61 ± 218.01	0.000 *	114.01	217.22
**Harris–Benedict**	0.00	67.75 ± 218.66	0.049	0.46	135.05	0.33	252.62 ± 207.97	0.000 *	203.40	301.85
**Owen**	0.15	328.22 **	0.000 *	192.37 ^Q1^	390.65 ^Q3^	0.22	374.38	0.000 *	325.55	423.20
**WHO/Schoefield**	0.09	292.97 ± 181.63	0.000 *	237.07	348.87	0.34	253.75 ± 201.42	0.000 *	206.07	301.42
**de Lorenzo**	0.18	(−)60.69 ± 168.77	0.023	−112.63	−8.75	0.23	113.11 ± 221.72	0.000 *	60.63	165.59
**Roza–Shigal**	0.13	178.63 ± 157.06	0.000 *	130.29	226.97	0.33	252.62 ± 207.97	0.000 *	203.40	301.85
**Bernstein**	0.08	428.53 ± 152.86	0.000 *	381.48	475.57	0.30	588.24 ± 215.83	0.000 *	537.15	639.32
**Tinsley FFM**	0.20	108.4 ± 165.68	0.000 *	57.41	159.38	0.30	146.48 ± 225.91	0.000 *	93.01	199.95
**Tinsley BW**	0.15	111.77 ± 202.05	0.000 *	49.58	173.94	0.34	186.09 ± 235.97	0.000 *	130.24	241.94
**ten Haaf FFM**	0.18	46.73 ± 160.5	0.063	−2.67	96.12	0.28	134.94 ± 219.39	0.000 *	83.01	186.87
**ten Haaf BW**	0.15	114.19 **	0.000 *	−42.15 ^Q1^	194.16 ^Q3^	0.33	109.55 ± 207.28	0.000 *	60.49	158.61

* *p* < 0,001; ** Wilcoxon test/lack of normal distribution, median presented; ^Q1^—first quartile, ^Q3^—third quartile; FFM—fat-free mass based; BW—body weight based; ICC—interclass coefficient; CI—confidence interval.

## Data Availability

The data presented in this study are available on request from the corresponding author. The data are not publicly available, because of privacy restrictions.

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
