# Peer review of "Accuracy of Resting Metabolic Rate Prediction Equations in Sport Climbers"

_ijerph, 2023, doi:10.3390/ijerph20054216_

Round 1

Reviewer 1 Report

Dear Editor,

thanks so much for the opportunity to revise the work entitled " Accuracy of resting metabolic rate prediction equations in sport climbers”.

The work is very interesting, showing the validation of well known equations for the estimation of Resting Metabolic Rate (RMR) in adult subjects. The work compares the accuracy of RMR calculated with 14 predictive equations with RMR measured by IC and try to identify the predictive equation with the highest correlation with IC measurement results.

The paper is well written, the results clearly reported and the statistical methods rigorous.

I suggest improving the discussion. Authors concluded that future researches are needed to develop sport specific RMR prediction equations. I get the feeling that the gap in literature wasn’t filled.  I missed authors’ suggestions how to create new equations to account for large differences in body composition and training levels, commonly observed in different athletes.

Author Response

Thank you very much for reading and reviewing my work. It was a pleasure to get feedback about the manuscript submitted. I found all of your comments very inspirational and I’m sure introducing the changes following your suggestions will add value to the manuscript.

Following your suggestions, I added the information about the criteria that should be taken into consideration while developing predictive equations for athletes of this type of discipline and highlighted the difficulties that this process brings.

Reviewer 2 Report

A sample

the large range in the age of the subjects should be commented, it affects the RMR.

Body mass affects RMR, climbers are expected to have a low body mass and a low percentage of adipose tissue, but not in this sample. That should be commented.

It would be good to add another sentence regarding the quality of the sample. Everything related to quality is:

The criterion for inclusion in the study was at least one year of regular sport climbing. The exclusion criterion was recreational climbing with less than once-a-week climbing practice.

There are certainly beginners in the sample. The question is are there top climbers in the sample?

The range of fat tissue is 5.5% min to 31.5% max. 5.5% can be experienced by top climbers and 31% by obese untrained people.

Table 2

Contraction force, blood pressure and pulse.

I assume that you needed these measurements to calculate RMR, but it is not clear from the text. It should be clarified in chapter Study protocol

Participants 80

were asked to indicate the highest grade, lead, or boulder, they had managed to redpoint

on three different routes/problems, on either an artificial wall or on a rock. The grade was

then standardized according to the International Rock-Climbing Research Association

(IRCRA) scale [14].

What happened next with participants climbing grade? If there are no further analyses with this grade it should be deleted.

You mention non-normal distribution. Which test you used to assess normality, add it to the chapter statistical analysis.

The sample includes climbers who train twice a week for more than a year. This is not a very representative group. I think there should have been stricter criteria? The sample varies significantly different in age, body composition, quality and experience. This should be stated in the limitations of the study, and

Author Response

Thank you very much for reading and reviewing my work. It was a pleasure to get feedback about the manuscript submitted. I found all of your comments very inspirational and I’m sure introducing the changes following your suggestions will add value to the manuscript.

Following your suggestions, I clarified some issues in the manuscript.

  1. Sample- it was surely the limitation of the study, I added the paragraph in the discussion comparing the characteristic of the group to another study involving climbers, and I pointed out this limitation of a wide variety of climbing grades and anthropometric parameters in the limitations section. I hope it was enough to make the quality of the sample visible and clear in the text.
  2. I added the information about the reason to collect blood pressure and heart rate before undertaking the IC test
  3. I filled in the information about the reasons to collect the information about the climbing grade as it was necessary to know the characteristic of the group.
  4. The statistic analysis section was supplemented with the information about the distribution check test.
  5. The limitation section was added and I did my best to show the limits of the study protocol and sample quality

Thank you very much again for the comments and suggestions, I hope that after those changes in the manuscript, the text will be clear for the readers.

Reviewer 3 Report

I'm starting from the minor issue:

- There are some minor typo error at line 193, 200 and 216 (I highlighted them in the pdf);

- The Bland–Altman plots are too small and not centered.

What I consider of medium entity are:

- There isn't a declaration of the limitation of the paper;

- In "Participants: The exclusion criterion was recreational climbing with less than once-a-week climbing practice." 
How do you know if the participants are telling you the truth about the frequency of their climbing practice?

- It confuses that "The exclusion criterion was recreational climbing with less than once-a-week climbing practice."
But in the Discussion is written: "We found significant differences between the measurements after dividing the 227 study group into recreational and sub-elite climbers, which may suggest that dividing 228 the study group according to climbing sophistication would help obtain more precise 229 results."

Were recreational climbers not left out? Think you have to justify this part

Author Response

Thank you very much for reading and reviewing my work. It was a pleasure to get feedback about the manuscript submitted. I found all of your comments very inspirational and I’m sure introducing the changes following your suggestions will add value to the manuscript.
1.    I corrected mistakes in the text and centered figures- I hope that in the published version it will be possible to add Bland Altman plots with the link to a better visible version.
2.     Sample- it was surely the limitation of the study, I added the paragraph in the discussion comparing the characteristic of the group to another study involving climbers, and I pointed out this limitation of a wide variety of climbing grades and anthropometric parameters in the limitations section and materials and methods section. I hope it was enough to make the quality of the sample visible and clear in the text. 
3.     As self-reported climbing ability is a valid representation of actual climbing ability I added this information in the text followed with reference.
Thank you very much again for the comments and suggestions, I hope that after those changes in the manuscript, the text will be clear for the readers.

Round 2

Reviewer 2 Report

All changes have been made, paper can be published in present form.